# RL FOR REASONING
# BY ADAPTIVELY REVEALING RATIONALES

**Mohammad Hossein Amani**[1]     **Aryo Lotfi**[1,2]     **Nicolas Mario Baldwin**[1]

**Samy Bengio**[1,2]     **Mehrdad Farajtabar**[2]     **Emmanuel Abbe**[1,2]     **Robert West**[1]

[1]EPFL     [2]Apple

## ABSTRACT

Learning in the combinatorially large output space of sequence generation problems is challenging as providing expert demonstrations scales poorly with sequence length, and RL struggles with sparse rewards. Between dense demonstrations in supervised training and no demonstrations in reinforcement learning lies an underexplored regime: partial supervision. We ask whether some classes of sequence learning problems become efficiently learnable by exploiting this gap. We address this by introducing **adaptive backtracking (AdaBack)**, a per-sample curriculum learning algorithm that reveals a partial prefix of the target output. The supervision length is adjusted dynamically for each sample based on the model's past reward signal, allowing it to incrementally learn to complete reasoning chains by conditioning on correct partial solutions. We investigate this *intermediate regime between SFT and RL* and argue that per-sample curriculum learning is more than a trade-off between efficiency and generality—it can succeed in tasks with long sequences of latent dependencies where SFT and RL both fail to generalize. Using a synthetic task with latent parity constraints, we show that AdaBack reliably solves problems that are otherwise intractable. On three mathematical reasoning benchmarks, DeepScaleR, MATH, and GSM8k, we find that AdaBack enables models to solve problems that RL alone cannot, acquiring new reasoning capabilities through incremental exposure to partial solutions.

## 1 INTRODUCTION

The reasoning capabilities of Transformers (Vaswani et al., 2017) have been extensively studied across domains such as mathematics (Saxton et al., 2019; Cobbe et al., 2021; Hendrycks et al., 2021; Lewkowycz et al., 2022), algorithmic reasoning (Veličković et al., 2022), and code generation (Chen et al., 2021). These studies indicate that the reasoning performance of such models is significantly enhanced by training and inference mechanisms involving explicit reasoning traces or rationales, commonly known as scratchpad (Nye et al., 2021) or chain-of-thought (Wei et al., 2023).

However, acquiring large-scale, high-quality reasoning traces for specialized domains, such as mathematics, poses substantial challenges. Reinforcement learning (RL)-inspired methods have emerged as a promising solution to this challenge. By utilizing (potentially noisy) reward functions or verifiers (e.g., checking final answers to math problems), language models can generate novel reasoning traces, effectively using the model as an RL policy. For instance, the REINFORCE algorithm (Williams, 1992) has been employed by methods like STaR (Zelikman et al., 2022), generating multiple solutions per problem and selectively fine-tuning on the correct ones. More advanced methods utilize algorithms like Proximal Policy Optimization (PPO) (Schulman et al., 2017) and, more recently, Group Relative Policy Optimization (GRPO) (Shao et al., 2024).

Despite recent progress, RL-based approaches for structured reasoning tasks continue to face significant hurdles. The key challenge lies in exploration: as reasoning chains grow longer, the space of valid output sequences increases exponentially, while reward signals remain sparse and often binary. Consequently, the probability of sampling a correct solution through random exploration diminishes exponentially with sequence length. As a result, standard RL tends to reinforce reasoning paths that are already assigned non-negligible probability by the pretrained model. Empirical evaluations by

Havrilla et al. (2024) and Yue et al. (2025) support this observation, showing that RL fine-tuning primarily amplifies existing behaviors without substantially exploring novel solution trajectories.

These limitations motivate our investigation of a third regime: the space **between supervised fine-tuning (SFT) and RL**. We study whether breaking down long reasoning chains into partial rationales—and adaptively conditioning the model on only a prefix of the target solution during training—can make exploration tractable, and ultimately expand the class of problems that sequence models can learn to solve. This leads to our central research question:

> Can reinforcement learning, when guided by adaptive partial supervision, teach models genuinely new reasoning capabilities? Specifically, for sequences learning tasks of variable lengths, can it enable discovery of solutions that were previously exponentially (in sequence length) unlikely under the model's initial distribution?

To illustrate this clearly, consider a reasoning task consisting of $n$ sequential steps, each of which must be performed correctly for the task to succeed (e.g., proving a math theorem). For simplicity, assume the model performs each step correctly with a constant probability $p$. Naively attempting the entire task yields a success probability of $p^n$, meaning that positive reinforcement signals would be exponentially infrequent—on average, every $p^{-n}$ iterations—making standard RL impractical. To mitigate this, we propose adaptive guidance: initially, we reveal all but the final step of the solution from the training dataset, using RL solely for this final step. Consequently, the model receives feedback with probability $p$ every iteration. As performance improves, we progressively conceal more steps of the solution, training the model to complete increasingly larger segments, maintaining frequent positive feedback. Eventually, the entire reasoning chain is learned step-by-step, effectively transforming a complex search space with success probability $p^n$ into $n$ simpler sub-searches, each with success probability $\Theta(p)$. While this example includes simplifying assumptions, we empirically demonstrate this phenomenon's practicality on a parity task described in Section 2.2.

The scenario described above is simplified and idealized. In real-world problems, reasoning steps may not be clearly distinguishable, and datasets typically contain tasks with varying difficulties and solution lengths, making uniform partial reveals inefficient. To address these practical challenges, we propose a per-sample adaptive algorithm that dynamically adjusts the revealed portion of each solution based on its inherent difficulty. Specifically, we leverage the GRPO framework (Shao et al., 2024), which naturally estimates task difficulty by generating multiple rollouts per question and averaging the rewards. We term this method *adaptive backtracking (AdaBack)*, detailed further in Section 2.

Backtracking strategies have precedents in RL—for example, the method of Salimans and Chen (2018). In language modeling, R3 (Xi et al., 2024) applies a curriculum that progressively trains models to complete reasoning chains from earlier positions. Building on these ideas, we ask: how can such slicing be adapted per sample, guided by model performance at each iteration, without relying on a heuristically designed global curriculum?

**Contributions**  Our contributions are as follows:

- We propose **AdaBack**, a per-sample adaptive curriculum learning algorithm that dynamically adjusts the amount of supervision during RL training based on reward feedback. AdaBack enables each sample to progress at its own rate, eliminating the need for manual curriculum staging or handcrafted schedules, transitioning from full supervision to full exploration in a data-driven way.

- We show that AdaBack bridges the gap between SFT and RL, enabling learning in regimes where both fail. On a synthetic parity task with sparse rewards, we demonstrate a **separation result**: AdaBack reliably solves the task, while SFT, RL, and their combination all fail.

- On standard mathematical reasoning benchmarks (DeepScaleR, MATH, and, GSM8k), we show that **AdaBack improves performance over standard RL and SFT+RL pipelines**. We also show that AdaBack applied to base models often matches the performance of SFT-initialized counterparts.

- We introduce further two variants of GSM8k: **Base-7**, which uses an unfamiliar numerical format never seen by the model during pretraining, and **Tensor-2**, which concatenates problems to increase reasoning depth. AdaBack achieves strong performance on both, demonstrating **robust generalization to symbolic shifts and longer-horizon reasoning**.

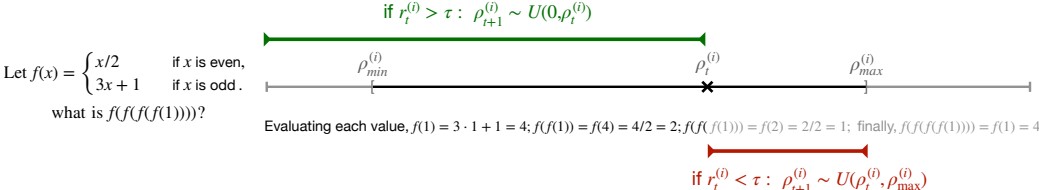

Figure 1: **AdaBack Update Rule.** At epoch $t$, we sample a supervision ratio $\rho_t^{(i)} \sim U(\rho_{\min}^{(i)}, \rho_{\max}^{(i)})$ and condition the model's generation on the question and the corresponding partial answer (shown in black text, with unrevealed content grayed out). If the average reward is below a threshold $\tau$, we increase supervision in the next epoch by sampling from the red interval $\rho_{t+1}^{(i)} \sim U(\rho_t^{(i)}, \rho_{\max}^{(i)})$. Otherwise, if the reward exceeds $\tau$, we reduce supervision and sample from $U(0, \rho_t^{(i)})$, the green interval, to make the task harder.

- Finally, we identify a limitation: for instruct-tuned models or models where pretraining has already exposed the model to most problem types, AdaBack together with standard RL training provide no benefits, underscoring their limitation in aiding exploration where uncertainty is low.

## 2 ADAPTIVE BACKTRACKING

The core idea is to expose a prefix of the target sequence during training and gradually reduce this supervision based on model performance. Unlike fixed-step or handcrafted curricula, this method allows each sample to progress at its own pace, naturally balancing difficulty and learning progress.

For RL algorithms like GRPO (Shao et al., 2024) or RLOO (Ahmadian et al., 2024) that perform multiple rollouts for each sample, we dynamically adjust the supervision level so that the average sample reward across the rollouts stays close to a desired amount (for example, 50% of the answers are correct for each math question). For other RL algorithms, the criteria to change the supervision level can depend on other estimation of the state's value.

**Problem Setup**  We denote random variables using capital letters. Let an input sample be represented as a sequence of tokens $X = (X_1, X_2, \ldots, X_\ell)$, corresponding to a natural language prompt, such as a question or problem description. The desired output is a chain-of-thought (CoT) style response $Y = (Y_1, Y_2, \ldots, Y_m)$, where each $Y_t$ represents a token or step in the model's reasoning process. We assume we have a dataset of such $(X, Y)$ pairs.

Our goal is to train a model to generate a correct reasoning sequence $Y$ conditioned on input $X$. Importantly, the correct CoT is not necessarily unique: multiple reasoning paths may lead to valid answers. However, we assume that correctness is verifiable via a reward model $r = R(X_{1:\ell}, \hat{Y}_{1:m})$ where $r = 1$ if the generated output $\hat{Y}$ is accepted as correct. We denote by $r_{\text{format}} \in [0, 1)$ the reward for a parsable generation when the generation has the correct format, and assume that $r = 0$ otherwise. This setup reflects many real-world reasoning tasks. For instance, in mathematical problem solving, final answers can often be verified, and different CoT traces may yield the same final result.

### 2.1 METHOD

At each training step, given an input $X^{(i)} = (X_1^{(i)}, \ldots, X_{\ell_i}^{(i)})$ and its corresponding target output $Y^{(i)} = (Y_1^{(i)}, \ldots, Y_{m_i}^{(i)})$, we reveal the first $k$ tokens of $Y$, $Y_{1:k}$, where $k = \lfloor \rho^{(i)} \cdot m_i \rfloor$ and $\rho^{(i)} \in [0, 1]$ is a sample-specific supervision portion. The model is trained to continue generation conditioned on the input and the revealed prefix $\hat{Y}_{k+1:m_i'}^{(i)} \sim P_\theta(\cdot \mid X^{(i)}, Y_{1:k}^{(i)})$, where $m_i' - k$ is the length of the generated continuation.

**Adaptive Update Rule**  For each training sample $i$, we maintain an interval $[\rho_{\min}^{(i)}, \rho_{\max}^{(i)}]$ initialized as $[0, 1]$, from which we uniformly sample the supervision portion $\rho_t^{(i)}$ for sample $i$ at epoch $t$. After generating a set of predictions and receiving an average reward $r_t^{(i)}$ (or other estimations of the state's

value if not using GRPO), we update this interval based on a fixed reward threshold $\tau$:

$$\text{If } r_t^{(i)} < \tau: \quad \rho_{\min}^{(i)} \leftarrow \rho_t^{(i)}$$
$$\text{If } r_t^{(i)} \geq \tau: \quad \rho_{\max}^{(i)} \leftarrow \rho_t^{(i)}, \quad \rho_{\min}^{(i)} \leftarrow 0.0$$

The next level of supervision is then uniformly sampled from this updated range:

$$\rho_{t+1}^{(i)} \sim U(\rho_{\min}^{(i)}, \rho_{\max}^{(i)}).$$

This process is visualized in Figure 1.

Intuitively, the model receives less supervision when it performs well and more when it struggles, enabling an automatic per-sample curriculum that adapts to the model's learning progress. This procedure naturally drives the model toward completing longer and more challenging portions of the target sequence, only when it shows competence on easier prefixes. The threshold $\tau$ governs how strictly we evaluate model success: higher values require stronger reward signals (e.g., higher average rewards) for the supervision ratio to decrease. In general terms, this process performs a form of stochastic binary search over the supervision ratio $\rho$, using reward feedback as a success signal. The goal is to minimize the amount of revealed rationale while ensuring the model continues to receive useful reward signals. As the model improves, the supervision ratio naturally decreases. For samples with no reward history, we initialize $\rho^{(i)}$ from global moving averages $\bar{\rho}_{\min}$ and $\bar{\rho}_{\max}$, which are updated over time using exponential moving averages. Additionally, to close train-test distribution mismatch, with a small probability we randomly set the portion to zero. These training details are further discussed in Appendix C.

This adaptive scheme enables each training sample to **follow its own trajectory from full supervision to full generation,** providing a flexible and data-efficient approach to curriculum learning in structured sequence tasks. This per-sample scheduling ensures that each training point advances only when ready, allowing the model to incrementally acquire reasoning skills without overfitting to fixed patterns.

## 2.2 CHAIN-OF-PARITIES: A SYNTHETIC ENVIRONMENT FOR STUDYING REASONING

To better understand learning dynamics in isolation from the complexities of natural language and pretraining, we introduce a synthetic sequence modeling task called the **chain-of-parities**. This task could be viewed as a **contextual blind cliff walk**, inspired by Schaul et al. (2015), adapted to reflect the challenges of chain-of-thought reasoning.

We aim to address a fundamental question: *Are there sequence learning tasks that cannot be learned by SFT, RL, or their naive combination but that can be solved by Adaptive Backtracking?* We construct such a task and demonstrate that the answer is affirmative.

Given a binary input sequence $X \in \{0,1\}^L$, the goal is to generate an output sequence $Y_1, Z_1, \ldots, Y_L, Z_L$ of length $2L$, where

- $Y_i \in \{0,1\}$ is unconstrained (both values are acceptable),
- $Z_i$ is the parity of $X_i, Y_i$, and $Z_{i-1}$, i.e, $Z_i = Z_{i-1} \oplus Y_i \oplus X_i$, where $\oplus$ denotes the XOR function and $Z_0 = 0$.

This design enforces a latent, step-wise structure over the output. The $Y_i$ values act as a "scratch-pad"—arbitrary values that influence $Z_i$ through an accumulating parity computation. In essence, each $Z_i$ encodes the parity of the prefix $(X_1, \ldots, X_i, Y_1, \ldots, Y_i)$, making the correctness of $Z_i$ dependent on the accuracy of $Z_{i-1}$ and prior chain-of-thought (CoT) steps. This recursive dependency mirrors real-world CoT tasks, where early mistakes cascade through the solution. Although there are $2^L$ valid outputs per input due to unconstrained $Y_i$, generating one randomly has a probability of just $2^{-L}$—analogous to sparse-reward regimes common in reasoning tasks.

We consider learning this task from a dataset of $n$ uniformly sampled sequences $(X_i, Y_i)$. This setup highlights the inherent limitations of SFT and RL:

- **RL fails**: Rewards are sparse, and discovering a single valid output via random exploration becomes exponentially unlikely as the problem length grows.

- **SFT fails**: For small training sets, supervised training alone cannot learn this task. Specifically, the task involves learning $n-1$ parity functions of degree three, for which the theoretical sample complexity is well-studied (Abbe et al., 2023a; Kou et al., 2024). For example, in the statistical query (SQ) model (Kearns, 1998), weakly learning a degree-$k$ parity from $L$ input bits requires at least $\Omega(L^{k-1})$ samples. Consequently, regular fixed-size neural networks trained via stochastic gradient descent (SGD) cannot even weakly learn a degree-three parity from only $n = \tilde{O}(L)$[1] samples (Abbe et al., 2023a).

- **SFT + RL fails**: With a limited number of training samples, such as $n = \tilde{O}(L)$ discussed above, SFT does not provide weak learning. Therefore, the exploration of the model remains random, and rewards remain exponentially sparse, hindering meaningful learning through standard RL.

In contrast, curriculum learning with adaptive supervision succeeds in solving this task using a limited number of training samples. Suppose the reward threshold is fixed at $0.5$. The adaptive curriculum can initially present the input question $X$ along with almost the entire solution—namely $Y_1, \ldots, Z_{L-1}$—to the model. In this case, the model only needs to generate the final step, $Y_L, Z_L$. Assuming the model has learned the task format during pretraining, the probability of generating a valid final step is approximately $0.5$, in contrast to $2^{-L}$ when generating the entire sequence.

This setup enables the model to explore and learn the final reasoning step. Specifically, for each partial sequence $X_1, \ldots, X_L, Y_1, \ldots, Z_{L-1}$, the model produces two valid continuations: $Y_L, Z_L$ and $Y_L', Z_L'$, where $x' = x \oplus 1$ denotes the complement of $x$. This variation allows gradient-based learners to infer that $Y_L$ directly influences $Z_L$: flipping $Y_L$ changes $Z_L$ while all other inputs remain fixed. Consequently, learning $Z_L$ reduces to learning a degree-two parity function, as one input ($Y_L$) is already learned, which has lower sample complexity than learning the degree-three parity function. In particular, degree-two parities can be learned with $\tilde{\Omega}(L)$ samples (Glasgow, 2024; Kou et al., 2024). Once the model masters the final step, the curriculum is adjusted to reveal a shorter solution prefix (up to $Z_{L-2}$), requiring the model to generate $Y_{L-1}, Z_{L-1}, Y_L, Z_L$. Since it has already learned to generate $Y_L, Z_L$, the focus shifts to learning the $(L-1)$-th step. This process is repeated iteratively by gradually shortening the hint until the model learns the full sequence. This progression demonstrates that there exist sample-size regimes where standard SFT + RL fails, but AdaBack successfully learns the task.

We note that the argument above is simplified, as the actual learning complexity depends on the specific learning model used. We will discuss such subtleties in Appendix B. Here, we focus on the empirical evidence supporting this phenomenon. We consider the case of $L = 16$ with a training set size of $n = 1024$, using the Llama 3.2 1B model (Grattafiori et al., 2024). The reward function assigns a value of 1 to fully correct sequences and $0.1$ to sequences that are syntactically valid (i.e., correct number of bits).

We first perform SFT on the training set to help the model learn the task format. We then compare the performance of standard RL and AdaBack on the task. As shown in Figure 2 (left), AdaBack achieves substantial reward early in training and progressively reduces the portion of the solution that is revealed, leading to seamless learning of the task. In contrast, standard RL (Figure 2, right) fails to obtain meaningful rewards and does not learn the task.

This experiment highlights how AdaBack enables guided exploration via partially correct sequences, effectively expanding the class of learnable problems in structured sequence modeling. It also establishes a clear separation between SFT + standard RL and AdaBack: while the former struggles with sparse rewards, AdaBack offers a principled middle ground, avoiding both the limitations of full supervision and the exploration challenges of standard RL.

## 3 EXPERIMENTS

In this section, we evaluate AdaBack on real-world reasoning tasks. In Section 2.2, we showed that AdaBack enables models to solve problems that are intractable under standard supervised fine-tuning, reinforcement learning, or their combination. Our goal here is to investigate whether AdaBack improves generalization and reasoning ability beyond what standard RL pipelines achieve on real-world reasoning tasks.

---

[1]We use $\tilde{O}$ to hide the logarithmic factors.

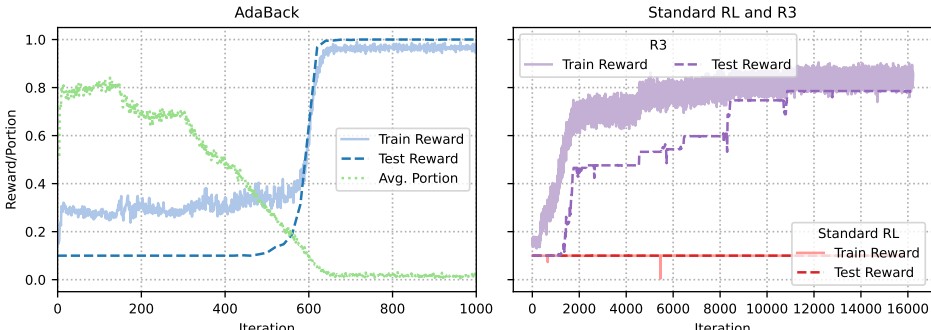

Figure 2: **Training Dynamics.** *Left:* Training and test rewards along with supervision ratios throughout training. With AdaBack, Llama 3.2 1B successfully learns the task in under 700 iterations. *Right:* Training and test rewards for SFT+RL (red) plateau at 0.1, indicating that only the output format—learned during supervised pretraining—has been retained. Test reward for R3 (Xi et al., 2024) is shown in purple; it reaches only 0.8 reward after more than 16,000 iterations. R3 segments training examples at all whitespace positions and applies RL uniformly over these fragments, resulting in inefficiency due to its non-adaptive strategy.

We train models using the GRPO algorithm (Shao et al., 2024) on base models of the Llama-3 family (Grattafiori et al., 2024). For fairness of attribution and clarity, we have followed the standard practice of works like (Shao et al., 2024; Xi et al., 2024) by using simple greedy or sampling-based decoding for our comparisons with a prompt as simple as 'Please provide the final answer to the question in \boxed{}.' All baseline experiments are ran until convergence.

The standard RL recipe involves first performing SFT on rationales, followed by RL fine-tuning (Gulcehre et al., 2023; Guan et al., 2025). However, this paradigm has recently been challenged by models trained solely with RL updates such as GRPO, without any SFT phase (DeepSeek-AI, 2025). Accordingly, we compare four variants: GRPO on a base model, GRPO on an SFT model, AdaBack-GRPO on a base model, and AdaBack-GRPO on an SFT model. Further experimental details are provided in Appendix D.

**Generalization on Natural Language Reasoning Tasks**   We evaluate on three standard mathematical reasoning benchmarks: DeepScaleR (Luo et al., 2025), MATH (Hendrycks et al., 2021) and GSM8k (Cobbe et al., 2021). In line with Mirzadeh et al. (2024) and Li et al. (2024), to assess generalization beyond pretraining exposure, we introduce two new variants of GSM8k:

- In **Base-7 GSM8k**, all numeric quantities and computations are represented in base-7.[2] While the problems themselves are unchanged, this symbolic shift forces the model to reason over a format it has not encountered during pretraining—analogous to deploying a model in a culture with a different numerical base. Performance on this dataset requires the model to generalize beyond familiar surface forms and reasoning chains observed in pretraining.
- We create **Tensor-2 GSM8k** inspired by Hosseini et al. (2024), by concatenating pairs of GSM8k problems and their solutions into single instances, yielding longer reasoning chains. While the resulting chains may be semantically disjoint, this construction increases the number of sequential reasoning steps required in the generation. This setup tests whether models can scale their reasoning over longer inputs and outputs, beyond what is required in the original GSM8k.

Table 1 summarizes performance across these tasks. We find that AdaBack consistently outperforms both GRPO and SFT+GRPO, particularly in out of pretraining distribution settings like Tensor-2 GSM8k. Notably, AdaBack applied directly to the base model often matches or exceeds the performance of the standard RL applied to the base and SFT-initialized models, suggesting that adaptive partial supervision on a base model may serve as a more effective prior than initializing from a SFT checkpoint. Interestingly, AdaBack applied on the base model may even outperform

---

[2]We dropped (fewer than 1000) samples where the question or answer required division, as base-10 divisions can have periodic representations in base-7.

AdaBack applied on the SFT model as in the Tensor-2 experiment. We elaborate on this phenomenon further in the text.

Table 1: Final test accuracy for each method across tasks and model sizes.

| Method | DeepScaleR | | MATH | | GSM8k | | Base-7 GSM8k | | Tensor-2 GSM8k | |
|---|---|---|---|---|---|---|---|---|---|---|
| | 1B | 3B | 1B | 3B | 1B | 3B | 1B | 3B | 1B | 3B |
| Base+RL | 6.8 | 6.6 | 6.4 | 15.0 | 7.9 | 63.7 | 4.8 | 4.9 | 0.0 | 0.0 |
| SFT+RL | 7.1 | 9.1 | 7.4 | 17.7 | 36.7 | 72.7 | 14.4 | 45.4 | 6.9 | 42.7 |
| AdaBack | 9.0 | 10.6 | 9.1 | 19.1 | 39.2 | **73.3** | 18.4 | 43.9 | 8.5 | **49.2** |
| SFT+AdaBack | **9.5** | **12.5** | **9.5** | **19.9** | **43.2** | 70.7 | **24.5** | **49.9** | **11.3** | 42.2 |

**Per-Sample Curriculum Without Manual Staging**   Curriculum learning typically requires hand-crafted stages, scheduling heuristics, and careful tuning of hyperparameters—such as how long to train at each difficulty level and with what parameters, and when to transition to harder ones. In contrast, AdaBack performs an automatic per-sample curriculum by adjusting supervision based on a simple reward threshold. Each training point progresses at its own pace, without global stage definitions. Although our implementation uses a stochastic binary search over supervision ratios, this is not essential: linear search or other adaptive strategies could be substituted. The key principle is to allow each example to progress at its own pace, without requiring global curriculum design or extensive hyperparameter tuning. As seen in Figure 3 (left) average portions naturally decrease and rewards increase with AdaBack without any boilerplate curriculum scaffolding. $\tau$ is the only hyperparameter needed to be set for AdaBack, and training is not sensitive to it. We have elaborated the choice of $\tau$ in Section C.

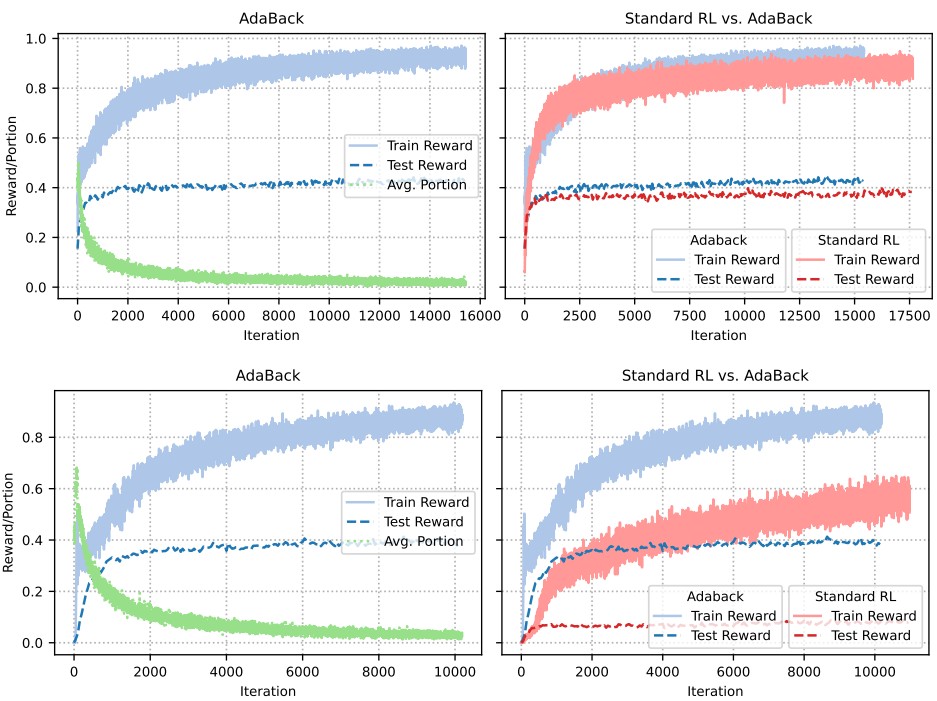

Figure 3: **AdaBack vs. Standard RL Across Model Initializations.** Results from training Llama3-1B on GSM8k dataset. The top row shows results for models initialized with SFT, while the bottom row shows base (non-SFT) models. The left column presents AdaBack training dynamics: train reward increases as supervision ratios (portions) decrease. The right column shows standard RL.

**Training Dynamics and Initialization**   Figure 3 shows training curves comparing AdaBack and standard RL across SFT and base model initializations. In both settings, AdaBack achieves better

rewards both at training and test compared to standard GRPO training. We provide additional training figures in Appendix F.

**Does AdaBack Expand the Model's Solution Space?** Yue et al. (2025) argue that RL fine-tuning reweights a model's output distribution without expanding its effective reasoning capacity. To test this claim, we evaluate models using **pass@k**[3], a metric that captures the breadth of plausible solutions. Figure 4 shows that AdaBack significantly improves pass@k over standard RL on both the base and SFT models, especially at large $k$. These gains are most pronounced when AdaBack is applied to the base model, again suggesting that it facilitates the discovery of new solution modes rather than just refining existing ones. If SFT-free RL with AdaBack introduces novel capabilities, we expect pass@k to increase even when base model coverage is low, which is what we show in Figure 4. In some cases, such as on the Tensor-2 GSM8k (see Table 1), AdaBack on the base model even outperforms AdaBack on the SFT-initialized model, suggesting that SFT may sometimes restrict the search space too early, limiting the benefits of exploration.[4] The latter can be observed in Figure 4 as well.

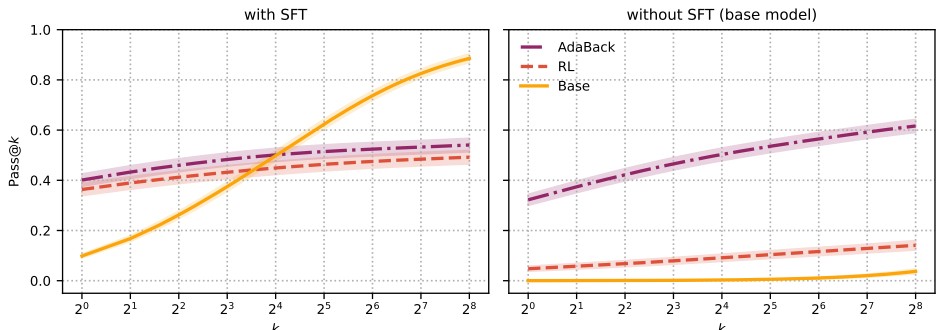

Figure 4: **Pass@k** for Llama3-1B SFT-initialized models (left) and base models (right) on GSM8k. AdaBack keeps a significant gap compared to standard RL and improves performance at higher $k$ even without SFT suggesting it expands the solution distribution rather than reweighting known answers (contra Yue et al. (2025)).

**When Does AdaBack Fail to Help?** On the MATH dataset with Llama 3.2 3B-Instruct, we observe no gains from AdaBack. As shown in Figure 10, both train and test rewards saturate quickly, with nearly all questions solved within a few hundred iterations. In Figure 11 we illustrate the training and test reward/accuracy trajectories for Qwen2.5-1.5B (both base and instruct) (Yang et al., 2024) on the GSM8k dataset, where we observed a similar behavior. We observed the same phenomena in the case of GSM8k for the 3B model in Table 1. This happens when RL reaches perfect training reward within a few iterations during training, i.e., when the dataset is too simple for the model. This suggests that these models have likely encountered much of the dataset during pretraining, leaving little room for RL improvements or for AdaBack to assist exploration. In such cases, AdaBack provides limited benefit—highlighting that its main strength lies in tasks where sparse reward or symbolic mismatch creates a significant learning barrier.

**Comparison with R3 on GSM8k, MATH, and DeepScaleR** Table 2 reports a direct comparison between AdaBack and R3 on three datasets of increasing difficulty—GSM8k < MATH < DeepScaleR (Luo et al., 2025)—using Llama-3 1B and Llama-3 3B base models. Across all settings, AdaBack and R3 rely on the same ground-truth solutions and reward functions; the only difference is the curriculum mechanism. This isolates the effect of adaptive per-sample supervision from the mixture approach of R3.

Overall, we observe a clear trend aligned with dataset difficulty. On DeepScaleR, AdaBack outperforms R3 for both 1B and 3B models. On MATH, AdaBack has an edge for the 1B model, though

---

[3]Pass@k is the probability that at least one out of $k$ model-generated outputs correctly solves a given problem.Chen et al. (2021); Brown et al. (2024)

[4]In an extreme case, the model may just memorize the answers during the SFT phase, removing hope for any exploration.

this edge diminishes for the larger 3B model. On GSM8k, AdaBack performs slightly below R3, though the absolute gap remains small relative to the inherent variance of RL post-training.

We believe two structural factors underlie this pattern:

**1. Segmentation brittleness on complex reasoning traces.** MATH and DeepScaleR contain long, multi-line reasoning steps, LaTeX/MathJax blocks, and irregular formatting. These properties make R3's slicing into fixed "stages" brittle and sensitive to heuristic delimiter choices, whereas AdaBack eliminates the need for any segmentation hyperparameters by using reward-driven adaptivity. On the other hand, for GSM8k, different steps of the solution can be easily separated using the newline delimiter. The latter can explain he superior performance of R3 and suggests that slicing solutions based on individual reasoning steps (as in R3) may work better than slicing the solution at random points (as done by AdaBack) when clean segmentation is possible. However, as discussed earlier, separating reasoning steps is not always feasible (e.g., in MATH and DeepScaleR).

**2. Per-sample adaptivity becomes increasingly important on heterogeneous, long-horizon tasks.** Harder datasets such as MATH and DeepScaleR contain many questions of widely varying structure and solution length. R3 differs from AdaBack in two aspects:

1. It slices all solutions into an equal number of pieces (e.g., 5), which is set globally for the whole dataset. Such global slicing does not account for variations in question difficulty and cannot adapt to intra-dataset heterogeneity. AdaBack's per-sample curriculum, which adjusts the supervision length independently for each example, is particularly advantageous in these settings and supports more stable progression on difficult problems.

2. Instead of imposing a curriculum over the data, R3 creates a mixture from all questions and sliced solutions. This contrasts with AdaBack's easy-to-hard curriculum. The use of a curriculum becomes increasingly important as the difficulty of the training set increases. In particular, Abbe et al. (2023b) provide theoretical results comparing curriculum learning and data mixtures: curriculum learning is beneficial when *easy* samples within a distribution are rare; conversely, if such samples are prevalent, standard training on the mixture may be more efficient than a curriculum.

To further support our argument, we refer the reader back to the chain-of-parities experiment presented in Figure 2, where it is shown that as the number of (non-trivial) reasoning steps in a task increases, the AdaBack approach can significantly outperform R3 (making training $> 20\times$ more efficient in Figure 2). The real-world datasets used in our paper still contain only a few reasoning steps. We believe AdaBack will better demonstrate its advantages on longer-horizon reasoning tasks (e.g., math Olympiad problems). However, we point out that experimenting in such regimes presents major challenges regarding data acquisition and large-scale compute.

Table 2: Comparison of R3 and AdaBack on GSM8k, MATH, and DeepScaleR

| Method | GSM8k | | MATH | | DeepScaleR | |
|---|---|---|---|---|---|---|
| | 1B (base) | 3B (base) | 1B (base) | 3B (base) | 1B (base) | 3B (base) |
| R3 | **41.5** | **74.2** | 7.8 | **19.2** | 6.6 | 9.6 |
| AdaBack | 39.2 | 73.3 | **9.1** | 19.1 | **9.0** | **10.6** |

## 4 RELATED WORK

We also provide a more extensive literature review in Appendix A.

**Limits of Reinforcement Learning for Reasoning**   Several studies have highlighted the difficulty of applying reinforcement learning (RL) to structured reasoning tasks. Exploration remains a major bottleneck: reward signals are sparse, and correct reasoning chains are exponentially rare in the output space. Havrilla et al. (2024) empirically evaluated several RL strategies on reasoning benchmarks and found that models fail to discover solutions outside the support of the base SFT model. They tested curriculum-inspired techniques such as *backtracking*—starting the model partway through a solution and gradually shifting the start point earlier Salimans and Chen (2018)—and *prioritized*

*level replay (PLR)*—sampling more frequently from difficult problems Jiang et al. (2020). Despite these interventions, performance gains were negligible: RL primarily amplified answers already assigned non-trivial probability by the pretrained model. These findings align with recent theoretical arguments by Yue et al. (2025), who claim that RL fine-tuning reweighs existing reasoning paths but fails to induce fundamentally new capabilities.

In contrast, we find that curriculum learning with partial supervision can induce such capabilities. Through carefully designed adaptive exposure to partial solutions, our models learn to complete problems that neither SFT nor RL could solve—even partially—highlighting the limitations of standard RL exploration and the promise of fine-grained curricula.

**Curriculum Learning** Curriculum learning was originally introduced to improve generalization and convergence by training models on simpler examples before harder ones (Bengio et al., 2009). In RL, backtracking (Salimans and Chen, 2018) and RFCL Tao et al. (2024) adopt this principle by initializing rollouts from later points in expert demonstrations and gradually shifting the start point backward.

In language modeling, R3 (Xi et al., 2024) introduces a step-wise curriculum where the model is trained to complete reasoning chains from progressively earlier points. However, despite being motivated as a dynamic curriculum, R3 effectively performs static data augmentation: demonstrations are sliced at delimiter positions (e.g., newlines in GSM8k), and each slice is treated as an independent training sample. In datasets without consistent delimiters, R3 falls back on uniformly partitioning each solution into a fixed number of segments (e.g., five), regardless of problem difficulty or solution length, and then mixing all segments across the dataset. This reliance on global heuristics limits scalability and applicability to domains without clear step boundaries, such as MATH (Hendrycks et al., 2021) or our synthetic parity benchmark. In particular, for the parity task we show that R3 has significantly worse learning speed and generalization in Figure 2. This is consistent with the theoretical results of Cornacchia and Mossel (2023); Abbe et al. (2023b) where it is proven that applying a curriculum over distributions can significantly improve the speed of convergence over simply mixing the distributions.

Our approach differs in both granularity and adaptability. We introduce an adaptive per-sample curriculum that dynamically controls the fraction of the output revealed to the model during training. Unlike R3, we do not rely on external step segmentation or dataset-specific structure. This fine-grained supervision allows the model to incrementally learn reasoning behaviors directly from reward signals, without explicit hinting or hard-coding intermediate targets.

## 5 CONCLUSION, LIMITATIONS, AND FUTURE WORK

Curriculum learning is often seen as a training heuristic to improve convergence. In this work, we show that per-sample curriculum learning via adaptive partial supervision enables fundamentally new reasoning capabilities that are inaccessible to standard supervised fine-tuning or reinforcement learning alone. Our proposed method, AdaBack, succeeds on a synthetic parity task where both RL and SFT fail, offering a constructive separation result. On real-world datasets such as MATH (Hendrycks et al., 2021), GSM8k (Cobbe et al., 2021), and its variants, AdaBack outperforms baseline RL and SFT+RL pipelines, often matching or exceeding standard RL applied on SFT-initialized models even when applied directly to a base model.

These results suggest that AdaBack not only improves sample efficiency, but also facilitates exploration of new solution modes, as evidenced by consistent gains in pass@k. However, we also observe that AdaBack provides limited benefit when the base model has already memorized much of the task, as appears to be the case with modern reasoning-apt models and math datasets.

A key limitation of our current implementation arises in the large dataset regime. When the number of unique training samples is high, most examples are seen infrequently and therefore rely heavily on global moving averages for supervision scheduling. This may reduce the effectiveness of per-sample adaptation. One promising direction for future work is to define supervision schedules not per sample, but per region in embedding space—e.g., using the average supervision level of the $k$ nearest neighbors of each sample. This would preserve the benefits of adaptation while remaining scalable in data-rich settings.

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

## A  EXTENDED RELATED WORK

**Scratchpads and Chain-of-Thought Strategies**   It is widely believed that success on challenging problems requires a model to know how to use intermediary computations in the context, to reason and deduct the final answers. Nye et al. (2021) proposed supervised training of Transformers to use scratchpads in addition to final answer, showing improvements on tasks like executing Python code and evaluating polynomials. Similarly, Wei et al. (2023) proposed chain-of-thought prompting, showing that large language models can generate scratchpads via in-context demos and without explicit training. Moreover, Kojima et al. (2022) studied zero-shot chain-of-thought generation for language models. Lanchantin et al. (2024) introduced the concept of self-notes, showing benefits of interleaving the intermediate reasoning steps within the question/context. Goyal et al. (2024) introduced pause tokens which act as place-holder tokens providing models with more computation time before output generations.

Several works have shown that allowing transformers to produce a chain-of-thought would increase their expressivity (Feng et al., 2024; Merrill and Sabharwal, 2024). Further, Abbe et al. (2024) put forward the notion of globality degree of a task as a hardness measure and show that scratchpads can make learning more efficient by breaking the globality of a task. They also proposed inductive scratchpads which impose a Markovian structure over reasoning steps, improving length generalization. Gao et al. (2025) proposed AbstRaL, showing improved robustness by training language models on mathematical abstraction of reasoning traces, instead of natural language chain-of-thoughts. Outside of reasoning and natural language scratch pads, unsupervised learning of intermediary symbolic sequences using straight-through gradient estimators (Bengio et al., 2013) has been studied in (Amani et al., 2024; Sánchez et al., 2023; Kaiser and Bengio, 2018).

**RL and Reasoning Benchmarks**   Reinforcement learning has emerged as a dominant paradigm for post-training language models on tasks involving sparse, verifiable reasoning signals Havrilla et al. (2024). Our reasoning tasks and settings are closest to STaR (Zelikman et al., 2022), Quiet-STaR (Zelikman et al., 2024), and R-STaR-MATH (Guan et al., 2025), which apply RL to improve mathematical and symbolic reasoning through selective training on verified rationales.

In parallel, alignment with human preferences has also leveraged RL techniques—most prominently Proximal Policy Optimization (PPO (Schulman et al., 2017)) (Bai et al., 2022; Pang et al., 2024a). Recent efforts have explored variants of REINFORCE-style updates and reward model bootstrapping, often using iterative fine-tuning pipelines. Interestingly, multiple works have reported that iterative filtering and fine-tuning on correct completions can match or exceed PPO performance in some domains (Gulcehre et al., 2023; Ahmadian et al., 2024).

Two standard benchmarks for mathematical reasoning used in most of the just-mentioned work in RL for reasoning are GSM8k (Cobbe et al., 2021) and MATH (Hendrycks et al., 2021) datasets. GSM8k consists of grade-school arithmetic problems requiring multi-step solutions, often accompanied by natural language justifications. MATH contains higher-difficulty competition-style math problems with structured step-by-step solutions. Nevertheless, RL approaches can be applied to any task for which verifiers or other forms of reward functions are available, including coding (Chen et al., 2021) and formal math (Yang et al., 2023).

Alternatively to RL, some natural language questions can be translated into formal logic and answered using automated theorem provers—for instance, Olausson et al. (2023) maps questions into first-order logic using an LLM, then delegates inference to a symbolic prover. AbstRaL also solves GSM8k-style problems by first converting them into abstract symbolic equations and then using a solver (Gao et al., 2025).

**Process vs Outcome Supervision**   Uesato et al. (2022) compared process-based feedback (reward each correct intermediate step) with outcome-based feedback (reward only at the final answer) on GSM8k. They found final answer accuracy can be similar with outcome-only training, but the quality of the reasoning steps was much higher with process supervision–the process supervision reducing reasoning errors dramatically. This could be another motivation for backtracking methods as the model learns by reinforcing small steps conditioned on correct chain-of-thoughts.

**Curriculum Learning in Reinforcement Learning**    Several other works in reinforcement learning have explored curriculum strategies to overcome sparse reward challenges (Narvekar et al., 2020). Florensa et al. (2017) uses a reverse curriculum for training, where starting states become increasingly difficult during training. However, compared to Salimans and Chen (2018), these starting states do not come from demonstrations. Backplay (Resnick et al., 2022) demonstrated strong gains in environments like Pommerman and Atari by starting episodes near the goal state using a single demonstration and gradually moving the starting point backward, enabling the agent to outperform the suboptimal demonstrator. Prioritized Level Replay (PLR) (Jiang et al., 2020) focuses training on the hardest levels in procedural environments like Procgen(Cobbe et al., 2019) by adaptively replaying levels where the agent performs poorly. Sukhbaatar et al. (2018b) proposed an automatic curriculum where a teacher agent proposes increasingly difficult tasks for a learner agent, leading to emergent complex curriculum strategies without any hand-designed task progression. This was extended in Sukhbaatar et al. (2018a) to learn goal embeddings and reusable low-level policies through self-play.

Another subset of works in curriculum learning (Chen et al., 2025; Shi et al., 2025; Bae et al., 2025) primarily adjust dataset/batch-level curricula, i.e., selecting which questions to include in each RL batch. By contrast, AdaBack introduces a per-sample curriculum, adaptively revealing partial rationales while sampling questions uniformly. Thus these approaches are complementary: AdaBack can be combined with batch-level curricula but unlike these works which do not benefit from chain-of-thought (CoT) demonstrations, AdaBack uses CoT supervised data to further improve efficiency.

**Comparison with State-of-the-art Mathematical Reasoning Models**    Although recent leaderboard results show strong performance, they are not directly comparable to our work because they typically rely on much larger models, additional math-heavy pretraining datasets, or complex prompting and inference-time searching strategies Xue et al. (2024); Pang et al. (2024b); Liu et al. (2023). In contrast, our focus is on isolating the contribution of AdaBack itself by comparing it fairly against standard RL baselines under controlled settings. Importantly, AdaBack is not an alternative to these methods but a complementary training enhancement. It can be combined with larger models, post-training strategies, or prompting techniques like chain-of-thought prompting and majority voting Xue et al. (2024) without overhead. This makes AdaBack a foundational component that strengthens models from the ground up and can integrate seamlessly into more complex state-of-the-art pipelines Pang et al. (2024b).

## B    DISCUSSION ON LEARNING PARITIES

The problem of learning parity functions has been extensively studied in the theory of machine learning (Abbe and Sandon, 2020; 2023; Barak et al., 2022; Abbe et al., 2023a; Edelman et al., 2024; Glasgow, 2024). Typically, the task is defined as follows[5]: each bit in the input is sampled independently and uniformly from the Rademacher distribution $\mathrm{Rad}(1/2)$ (i.e., $-1$ or $1$ with equal probability). The target function is a parity over a subset of bits, i.e., $\prod_{i \in S} x_i$ for some $S \subseteq [d]$. The size of $S$, denoted $|S|$, is called the degree of the parity. When $|S| = O_d(1)$ is constant (with respect to the input dimension), the problem is referred to as learning a sparse parity.

Due to symmetry, we often discuss learning a parity of degree $k$ without specifying the particular subset $S$. Most learning formulations assume the degree $k$ is known while the identity of $S$ must be learned, yielding a hypothesis class of size $\binom{d}{k}$. It is well established that the difficulty of learning sparse parities increases with the degree $k$. In the statistical query (SQ) model (Kearns, 1998), it has been shown that learning a parity of degree $k$ requires $\Omega(d^k)$ queries, which translates to $\Omega(d^{k-1})$ samples (Kou et al., 2024).

Similar lower bounds have been conjectured/shown for neural networks. In particular, fully connected networks with bounded width and depth and rotationally invariant weight initializations are conjectured to require $\Omega(d^{\max(k-1,1)})$ online gradient steps to learn a degree-$k$ parity (Abbe et al., 2023a). This has been proven under specific assumptions, such as for noisy population gradient descent (Abbe and Sandon, 2023; Abbe et al., 2021). Recently, Kou et al. (2024) showed that this lower bound is

---

[5]We follow standard theoretical conventions and use $\pm 1$ values for the bits. Equivalent results hold for $0, 1$ bits. Moreover, since Transformers embed inputs into continuous vectors, empirical results are invariant to the bit representation.

tight: a variant of SGD (namely, sign-SGD) can learn the task in $O(\log d)$ iterations using batches of size $\tilde{O}(d^{k-1})$. The problem becomes more subtle in the offline setting where samples can be reused. In this case, Abbe et al. (2023a) conjecture the optimal sample complexity to be $\Theta(d^{\max(k/2,1)})$. We note that one can often reduce the sample complexity at the cost of using larger models, where the width/depth of the model scales with $d$ (Edelman et al., 2024).

In our proposed chain-of-parities task, an input sequence of length $L$ contains $L-1$ parity targets of degree 3 and one initial parity of degree 2. Learning all of these parity targets jointly with a single model may be harder (due to interference) or easier (due to shared information, such as the influence of previous bit), depending on the model architecture—particularly positional embeddings in Transformers. A simplifying assumption is to treat the learning of each parity target as independent. Under this assumption, the problem reduces to learning parities of degree 3, for which, based on the discussions above, one would not expect learnability using only $\tilde{\Theta}(L)$ samples. In contrast, the degree-2 parity should be learnable with $\tilde{\Theta}(L)$ samples (Glasgow, 2024; Kou et al., 2024).

This explains why supervised fine-tuning with $\tilde{\Theta}(L)$ samples fails to weakly learn the degree-3 parity targets $Z_i$ for $i > 1$. Consequently, SFT followed by standard reinforcement learning also fails, as the RL phase remains unguided which makes the probability of stumbling upon a valid completion (and receiving reward) exponentially small.

We now explain why AdaBack can succeed where SFT and RL fail. Consider a scenario in which the first $L-1$ steps of the solution, i.e., $Y_1, Z_1, \ldots, Y_{L-1}, Z_{L-1}$, are provided, and the model must complete $Y_L, Z_L$. In this setup, there are two valid completions: $(Y_L, Z_L)$ and its complement $(Y'_L, Z'_L)$, where $x'$ denotes the bitwise complement of $x$. The fact that the same prefix $X_1, \ldots, X_L, Y_1, \ldots, Z_{L-1}$ admits two such completions indicates that if $Z_L$ is a parity of the previous coordinates, then $Y_L$ lies in the support of this parity. This effectively reduces the problem of learning a degree-3 parity function for $Z_L$ to that of learning a degree-2 parity function (since one bit is already revealed)—a task that is learnable with $\tilde{\Theta}(L)$ samples. This introduces a setting where AdaBack is able to learn the task, while the combination of SFT and standard RL fails.

We do not explore the exact training dynamics here, as they depend on the model architecture and optimization details. Instead, we empirically validate this separation result in Sec. 2.2.

## C  METHOD DETAILS

**Improving Convergence via $\rho = 0$ Injection**   We observed that always sampling supervision ratios from $\rho_t^{(i)} \sim \mathcal{U}(\rho_{\min}^{(i)}, \rho_{\max}^{(i)})$ led to slower convergence and a distribution mismatch between training and validation. Specifically, when a large portion of samples remain difficult (i.e., $\rho_{\min}^{(i)} > c$ for many $i$), the model would be significantly less exposed to fully unsupervised completions during training, while validation is always conducted at $\rho = 0$.

To address this, we introduced a small amount of supervision-free training into the curriculum: with 10% probability, we sample $\rho_t^{(i)} = 0$ directly, while using the uniform interval sampling with the remaining 90% probability. This stochastic exposure to 0-portion training helped close the train-test gap and accelerated convergence in the early phases of AdaBack training.

**Bootstrapping via Global Averages**   For samples with no reward history, we initialize $\rho^{(i)}$ from global moving averages $\bar{\rho}_{\min}$ and $\bar{\rho}_{\max}$, which are updated over time using exponential moving averages:

$$\bar{\rho}_{\min} \leftarrow \alpha \rho_{\min}^{(i)} + (1 - \alpha) \cdot \bar{\rho}_{\min}$$
$$\bar{\rho}_{\max} \leftarrow \alpha \rho_{\max}^{(i)} + (1 - \alpha) \cdot \bar{\rho}_{\max}$$

Here $\alpha \in [0, 1]$ is a momentum parameter controlling the smoothing rate.

**Choice of the reward threshold $\tau$**   Curriculum learning often requires careful hyperparameter tuning—such as deciding how long to train at each stage and with what optimizer parameters or

learning rate. AdaBack simplifies this process by collapsing these decisions into a single parameter: the reward threshold $\tau$. Intuitively, $\tau$ controls the average reward the model receives; the binary search mechanism ensures that the supervision is adapted to maintain this target. We experimented with several values of $\tau$ and found the final results to be largely robust, except near extreme values (i.e., $\tau \approx 0$ or $\tau \approx 1$). We chose $\tau = 0.5$ in our main experiments, as this maximizes reward variance, following the insights of Razin et al. (2024) on avoiding vanishing gradients for PPO updates.

**Sensitivity to the Reward Threshold**  We evaluate AdaBack with reward threshold values $\tau \in \{0.1, 0.2, \ldots, 0.9\}$ for RL training of the finetuned Llama3.2-1B on GSM8k. Figures 5 and 6 report training reward, test accuracy, and the evolution of the average revealed portion over the course of approximately 12k RL iterations.

It can be observed that the learning dynamics is generally similar for all the considered thresholds. It is only for high thresholds such as $\tau = 0.8$ and $\tau = 0.9$ that a different behavior is observed at the beginning of training. These higher thresholds result in a higher supervision portion and train reward and slightly worse test reward at the beginning of training.[6] Nevertheless, as training continues, it can be observed that the final performance, train/test rewards, and portions converge to similar values, becoming essentially indistinguishable. This indicates that AdaBack is highly robust to the choice of $\tau$. Given this general robustness, we take the value $\tau = 0.5$ in our main experiments which aims to maximize the reward variance, consistent with the theoretical studies of Razin et al. (2024).

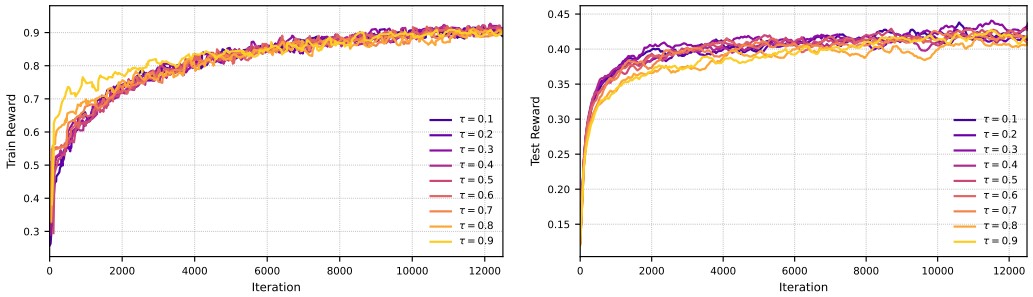

Figure 5: **Training reward (left) and test accuracy (right) across different AdaBack reward thresholds.** Although there is some difference at the begining of training for high thresholds ($\tau = 0.8$ and $\tau = 0.9$), the learning curves and performance become indistinguishable as training goes on.

## D   EXPERIMENT DETAILS

**Computation and Budget**  We conducted experiments using small language models ranging from 1B to 3B parameters. Input sequence lengths varied across tasks, with a maximum input length of 2048 tokens and generation lengths up to 2048 tokens. Depending on model size and sequence length, we used nodes equipped with either 4 or 8 NVIDIA A100 GPUs. Including all diagnostic and development runs, our total compute usage amounted to approximately 80,000 A100 GPU hours.

**Reinforcement Learning Setup**  We used the GRPO algorithm for all RL experiments, with 8 rollouts (generations) per input sample. Unless otherwise noted, we trained all models for at least 10,000 iterations to ensure convergence to long-term behavior. To further ensure the fairness of comparisons, we continued each standard GRPO training at least as long as the corresponding AdaBack training. Exploring different learning rates, we found $lr = 10^{-6}$ to give us stable and rather fast training. We report final test accuracy or reward as the average over the last 5 checkpoints. In rare cases where performance deteriorated at the end of training, we averaged over the last 5 iterations with increasing reward.

Our RL batch size was 256. We did not use a KL penalty or entropy regularization, neither in the loss nor in the reward.

---

[6]Note that these thresholds reveal a larger part of the answer at the beginning, hence, it is natural that the test reward is comparatively lower at the beginning.

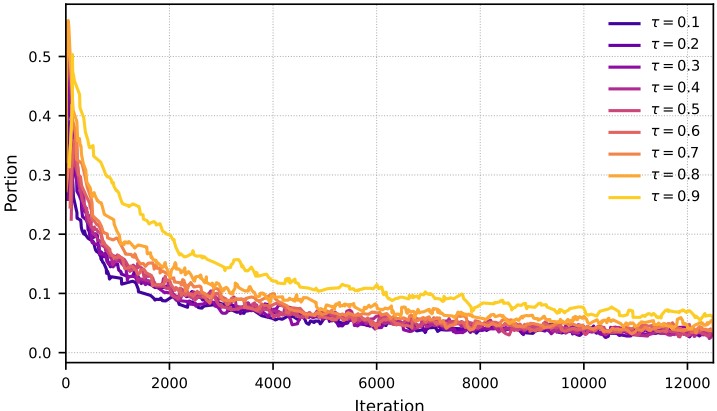

Figure 6: **Dynamics of the average revealed portion for different reward thresholds.** High thresholds such as $\tau = 0.8$ and $\tau = 0.9$ result in a larger average revealed portion in the initial part of training. Nonetheless, as training continues, all thresholds converge to stable supervision levels. Combined with Figure 5, this suggests AdaBack's curriculum adapts similarly across a wide range of thresholds.

For supervised-fine tuning of the base models, we used a validation set to adjust the hyperparameters and also the number of fine tuning iterations.

For datasets like Tensor-2 GSM8k and the chain-of-parities task, where the output format is nontrivial, we applied a format reward of $r_{\text{format}} = 0.1$ to encourage structured outputs. This was unnecessary for models initialized from SFT, which already generate syntactically valid sequences and thus begin training with an initial reward near 0.1. However, for consistency, we kept this term in all settings.

**Chain-of-Parities Task Setup** We use 1024 labeled examples with sequence length $L = 16$. In this setting, the probability of generating a valid solution by chance is $2^{-16}$, making reward signals extremely sparse. As a result, standard RL fails to make progress, even when initialized from an SFT checkpoint. The curriculum introduced by AdaBack mitigates this by incrementally revealing intermediate reasoning steps, allowing learning to proceed from the final step backward.

## E  LLM USAGE

We have used ChatGPT and Gemini models to improve text quality at phrase and paragraph level.

## F  ADDITIONAL FIGURES

We show how portions and the train and test rewards evolve for GSM8k, MATH, Base-7 GSM8k, and Tensor-2 GSM8k in Figures 3, 7, 8, and 9 respectively.

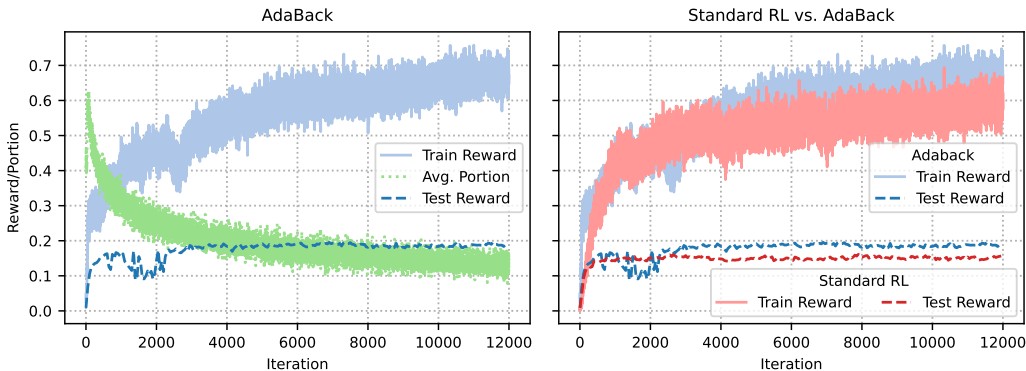

Figure 7: **AdaBack on MATH.** Results from training Llama3-3B base model (non-SFT) on MATH dataset. The left column presents AdaBack training dynamics and the right column shows standard RL for comparison.

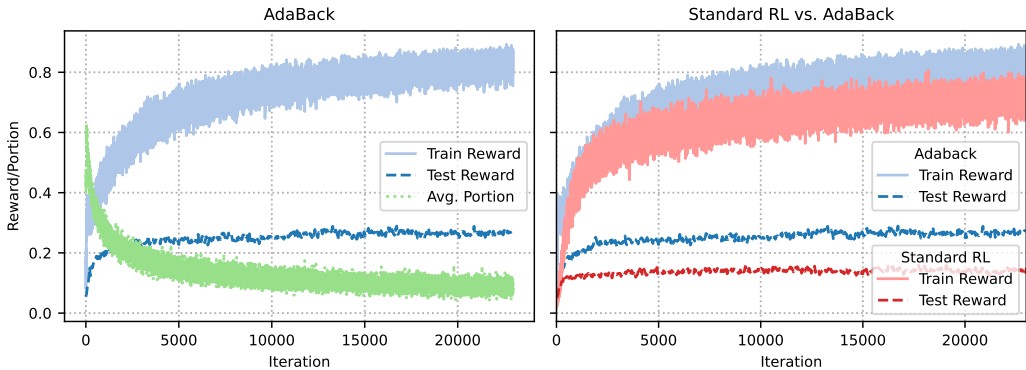

Figure 8: **AdaBack on Base-7 GSM8k.** Results from training Llama3-1B fine-tuned model (SFT) on Base-7 GSM8k dataset. The left column presents AdaBack training dynamics and the right column shows standard RL for comparison.

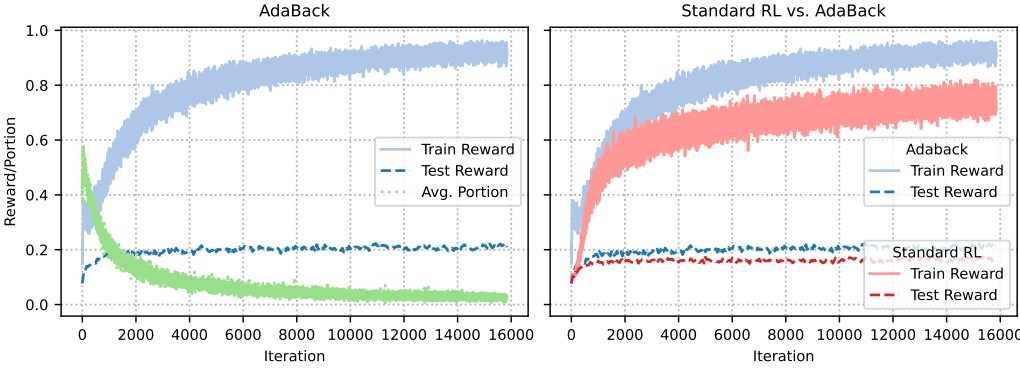

Figure 9: **AdaBack on Tensor-2 GSM8k.** Results from training Llama3-1B fine-tuned model (SFT) on Tensor-2 GSM8k dataset. The left column presents AdaBack training dynamics and the right column shows standard RL for comparison. Note that for this task, outputting an answer with the correct format has $0.1$ reward. So the observed rewards are higher than the actual accuracies.

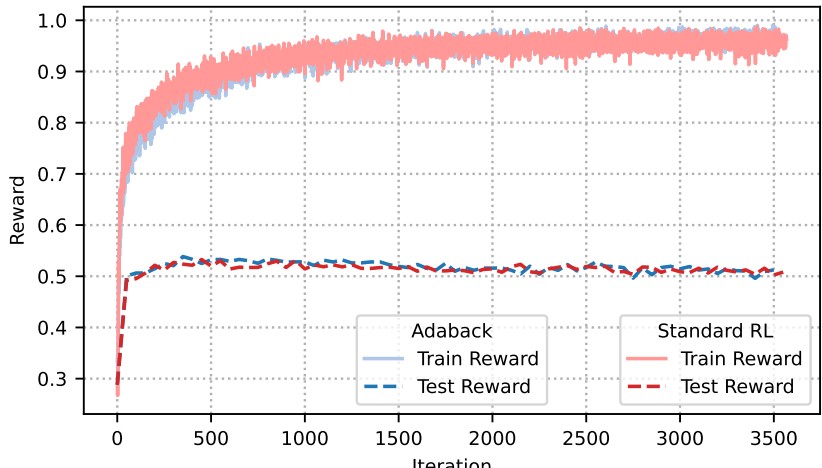

Figure 10: **Lack of Learning Signal on MATH.** On the MATH dataset, Llama 3.2 3B-Instruct shows minimal learning dynamics, with both train and test rewards saturating early. Nearly all questions are solved within a few hundred iterations, leaving no room for AdaBack to provide further benefit. This may be due to the model having been heavily exposed to examples similar to this dataset during pretraining.

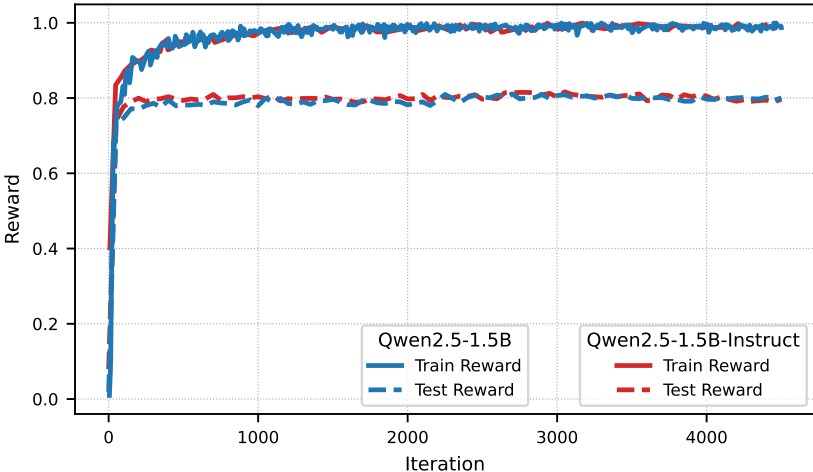

Figure 11: **Learning dynamics of Qwen2.5-1.5B models on GSM8k.** Training reward and test accuracy for both the base and instruct variants. Both models rapidly saturate the test reward **after only a few iterations**, exhibiting limited learning signal, mirroring the saturation effect previously reported for Llama-3.2-3B-Instruct (see Figure 10). This suggests that the dataset is effectively solved by these models during pretraining and provides limited learning signal for RL fine-tuning.

