# OpenReview forum: "RL for Reasoning by Adaptively Revealing Rationales"
_ICLR.cc/2026/Conference — ICLR 2026 Poster_

### Official Review · Reviewer_hbbC · 2025-10-31

**Soundness:** 3
**Presentation:** 3
**Contribution:** 2
**Rating:** 4
**Confidence:** 3

**Summary:**

The paper tackles the problem of RL for long-horizon, CoT reasoning, where exploration is hard because correct sequences are exponentially rare. It proposes AdaBack, a per-sample curriculum that, for each training example, reveals a variable-length prefix of the gold rationale and uses the reward signal to shrink or expand that revealed prefix over time, effectively doing a stochastic binary search on “how much to reveal.” The method is instantiated on top of GRPO and aims to occupy the underexplored middle ground between fully SFT on rationales and fully unsupervised RL. The authors show (i) on a synthetic chain-of-parities task that SFT, RL, and SFT+RL all fail, but AdaBack succeeds, and (ii) on MATH, GSM8k plus two harder GSM8k variants (Base-7 and Tensor-2) that AdaBack improves over RL and SFT+RL, sometimes even when starting from a base model rather than an SFT one (Table 1). They also identify a regime where AdaBack does not help -- when the base/instruct model has essentially memorized the dataset—arguing that AdaBack’s main value is in exploration-limited settings.

**Strengths:**

1. Originality: The paper identifies a concrete, underexplored regime “between SFT and RL” and operationalizes it as adaptive partial supervision that is per-sample, not global.
2. Experiments span standard math (MATH, GSM8k) and two carefully designed variants (Base-7 and Tensor-2) that stress symbolic shift and depth, and AdaBack improves or matches baselines in most cells of Table 1.
3. Fig. 4 suggest AdaBack expands the solution distribution, not just reweights existing solutions.

**Weaknesses:**

1. The benchmarks are too easy, especially GSM8k.
2. The key hyperparameter $\tau$ effectively sets how aggressively the supervision interval is shrunk/expanded, but there is no systematic sensitivity analysis (e.g., $\tau$=0.1, 0.3, 0.5, 0.7) to show robustness across tasks.
3. It seens like AdaBack is not that general. The authors focus on tasks that have a narrow scope.
4. The method keeps per-sample supervision intervals and falls back to global moving averages when samples are rarely seen; this is acknowledged as a limitation (Sec. 5), but the paper does not quantify how quickly performance degrades as dataset size grows. A synthetic experiment varying number of unique problems would clarify scalability.

**Questions:**

1. Could you conduct experiments on a more difficult benchmark such as AIME24 or AIME25?
2. You mention that AdaBack can be used with non-GRPO RL (via other value estimates). Did you try plain REINFORCE/RLOO, and if so, were the gains comparable?
3. Could you report variance / multiple seeds for the chain-of-parities experiment in Fig. 2?
4. How many rollouts per question were used in GRPO, and does AdaBack still help if we reduce that number (i.e., with weaker reward estimation)?
5. Could you give a systematic sensitivity analysis about the hyperparameter $\tau$?
6. In Table 1, SFT+AdaBack on GSM8k 3B is actually worse than AdaBack on the base model (70.7 vs 73.3). Can you clarify this?

---

> ### Author Response · Authors · 2025-11-20
>
> Thank you for your review and the questions. Please find our responses below.
>
> ### W1 / Q1. Benchmark simplicity and AIME
> Regarding AIME, the AIME24 set contains only 30 problems, insufficient to meaningfully study RL training dynamics or generalization. Earlier AIME years contain final answers only, without rationales, making them unsuitable for prefix-based curricula. In practice, the community typically trains on large reasoning datasets and evaluates zero-shot performance on AIME (e.g., [1]). It is, in principle, possible to curate this dataset, but one needs to crawl math problem solving forums and websites and check human generated proofs.
> If the reviewer is aware of high-quality reasoning datasets with reliable CoT annotations, we would be grateful for pointers and would run our experiments on them.
>
> ### W2, Q5. sensitivity to reward threshold.
> We discuss threshold sensitivity in lines 915–917. Across the range of values tested, we observed nearly identical final performance, which is why the main paper omits the plot. We will include the figure in the next revision for clarity.
>
> ### W3. Scope and generality of AdaBack
> AdaBack addresses the general problem of exploration under sparse rewards, an issue that arises widely in RL settings, not only in math. The mechanism requires only:
> 1. a dataset of question–answer pairs (with rationales), and
> 2. a verifiable reward function.
>
> Math datasets provide a clean environment where sparse-reward issues are pronounced and reasoning depth is measurable, which is why we focus our empirical study there. However, the method is not tied to mathematics; we expect it to apply to any domain where reward sparsity limits exploration.
> We would appreciate clarification from the reviewer regarding what aspects seem scope-limited, so we can better address the concern. We would like to finally emphasize that focusing RL for reasoning experiments on math datasets is actually a quite common practice in studies of RL for reasoning [1,2,3,4] ([1,2] train on one single math dataset).
>
>
> ### W4. Scalability and reliance on a global moving average
>
> As noted in Sec. 5, when datasets are extremely large and each question is seen very rarely, AdaBack falls back on the global moving averages. This regime is uncommon in current RL-for-reasoning pipelines, which typically perform multiple epochs even on large datasets. Under such training schedules, AdaBack continues to operate in its per-sample mode for most examples.
> A controlled synthetic experiment varying dataset size is a useful direction and we will consider it in future work. Unfortunately, constructing a large-scale dataset with clean human CoT remains a bottleneck: large open-source datasets such as OpenMathReasoning or OpenR1-Math-220k contain LLM-generated rationales, many of which are long, noisy, or unverifiable.
>
>
> ### Q2. AdaBack with REINFORCE or RLOO
>
> We have not run experiments with REINFORCE or RLOO. AdaBack requires a question-dependent value estimate, as the supervision ratio adapts to the estimated difficulty of each example.
> RLOO, like GRPO, provides Monte Carlo value estimates, and we expect AdaBack to exhibit similar benefits in this setting. REINFORCE uses one Monte Carlo sample to compute the updates, while one can use the reward for the same rollout to compute AdaBack portion update, this would be a noisier estimation compared to RLOO or GRPO.
> We focused on GRPO because it is the standard value estimator in recent reasoning pipelines and provides a stable, unbiased per-question advantage estimate. We agree that exploring additional estimators (e.g., PPO’s learned value function) is interesting future work; It would be interesting to see the synergy of AdaBack with other value estimators, but that will also multiply the already very high cost of experimentation.
>
>
> ### Q3. Multi-seed results on chain-of-parities
> We are happy to run additional seeds, however, could you clarify which metric you would like to see? In our initial experiments, the difference between AdaBack and baselines was qualitatively large: AdaBack solves the task in <1000 iterations (less than ~2 hours on one node), whereas R3 and RL fail to learn even after 16,000 iterations(> 1day of training.) Because of this stark separation, multi-seed runs were not included in the initial submission.
>
>
> ### Q4. Number of GRPO rollouts
> As stated in line 926, we use k = 8 rollouts per question. We did not vary k due to the additional cost of RL experiments. Notably, our rollout count lies on the lower side of what is commonly used: prior work uses 16 [1,2], 32 [3], or even higher values [4]. Testing AdaBack across different k values is a promising direction, but cost-intensive.

---

> > ### Author Response · Authors · 2025-11-20
> >
> > ### Q6. Why AdaBack sometimes performs better on base models than SFT models
> > As discussed in lines 318 and 382, our hypothesis is that SFT narrows the model’s exploration distribution. A strongly SFT-trained model may commit to a particular reasoning path, reducing entropy and limiting the ability of RL (and AdaBack) to discover new solution modes.
> > In contrast, a base model retains higher variability in early RL iterations, enabling AdaBack to more effectively expand the solution space, as also reflected in our pass@k analysis.
> > Understanding when this effect occurs is an interesting open question requiring larger-scale studies.
> >
> >
> > Please let us know if you have any other concerns.
> >
> > ### References
> > [1] The Art of Scaling Reinforcement Learning Compute for LLMs, Khatri et al., 2025
> >
> > [2] DAPO: An Open-Source LLM Reinforcement Learning System at Scale, Yu et al., 2025
> >
> > [3] Can Large Reasoning Models Self-Train?, Shafayat et al., 2025
> >
> > [4] Not all Rollouts are useful: Down-Sampling Rollouts in LLM Reinforcement Learning, Xu et al., 2025

---

> > > ### Comment · Reviewer_hbbC · 2025-11-27
> > >
> > > Thank you for the author's detailed response. I have another question: the author mentioned that the "mechanism requires a dataset of question–answer pairs (with rationales)," and it seems that the rationales need to be verified, such as "LLM-generated rationales, many of which are long, noisy, or unverifiable." Does this mean that AdaBack is somewhat more limited compared to the widely used SFT+RL approach?

---

> ### Author Response · Authors · 2025-11-27
>
> Dear Reviewer,
>
> Thank you for your follow-up question.
>
> **Regarding W2, Q5 (sensitivity to the reward threshold):**
> We have added a full threshold-sensitivity analysis in the updated appendix (final page of the new PDF). In summary, AdaBack behaves nearly identically for thresholds in the range $\tau \in [0.1, 0.9]$, with only extreme values ($0.8, 0.9$) showing mild deviation early in training. This confirms that AdaBack is robust to the choice of $\tau$.
>
> **Regarding your question on AdaBack vs. SFT+RL:** AdaBack does **not** require more or better data than standard SFT+RL. The requirements of the two approaches are essentially identical:
>
> - If a dataset contains question–answer pairs with rationales (as needed for SFT),
> - and if RL is applied to those questions (as in typical SFT $\rightarrow$ RL pipelines),
> - then the same dataset can be used for AdaBack—with or without the SFT step.
>
> These cases correspond exactly to the four configurations evaluated in Table 1.
>
> Our comment about noisy or unverifiable LLM-generated rationales applies equally to **all** methods that rely on gold or pseudo-gold rationales, including SFT and SFT+RL. If the rationales are unreliable, SFT may teach incorrect intermediate reasoning steps, and RL on top of such data may amplify those errors. In such settings, AdaBack faces the same limitations as SFT+RL not a stronger one.
>
> For this reason, we (and most recent RL-for-reasoning work) focus on high-quality datasets with human-written CoT such as MATH and GSM8K. Whenever high-quality rationales are available, AdaBack can be applied under the same assumptions as SFT+RL, while providing additional benefits in sparse-reward regimes.
>
> Please let us know if further clarification would be helpful.

---

### Official Review · Reviewer_Rx4E · 2025-11-01

**Soundness:** 3
**Presentation:** 3
**Contribution:** 3
**Rating:** 8
**Confidence:** 4

**Summary:**

Authors identify shortcomings in SFT and RL for sequential reasoning tasks:
- SFT: data collection is expensive and theoretically insufficient for certain symbolic tasks given insufficient training data
- RL: outcome rewards are sparse and the policy really gets meaningful signal when a successful trajectory is found which may be exponentially unlikely (in solution length)

Authors propose a middle ground between SFT and RL that uses RL training with problem contexts augmented with prefixes of SFT train data. Moreover, the proportion of the SFT solution revealed to the model in training is dynamically adjusted based on model performance.

Their method produces encouraging results on GSM8K and MATH.

**Strengths:**

Clear conceptual contribution:
- The paper identifies a meaningful regime between SFT and RL and proposes a principled mechanism (adaptive partial supervision) rather than a heuristic curriculum schedule. This framing is conceptually clean and well-motivated.

Per-sample adaptive curriculum is novel and well-justified:
- Unlike prior curriculum or backtracking approaches that rely on global schedules or coarse segmentation heuristics (e.g., R3), AdaBack adapts per-instance based on reward signals. This allows difficulty to scale automatically in heterogeneous datasets.

Empirical Analysis:
- The training dynamics figures, pass@k exploration analysis, and comparisons across both base and SFT-initialized models provide a clear and multi-angle understanding of how the method behaves. The results are presented in a way that makes the mechanism’s effects interpretable.

Promising results on reasoning benchmarks: On GSM8k (including the proposed Base-7 and Tensor-2 variants) and MATH, AdaBack produces consistent improvements over standard RL pipelines

**Weaknesses:**

- limited scale: the models explored in this work are limited to 3B parameters
  - The deepseek R1 report asserts that smaller models have lower capacity from benefiting from RL to discover novel reasoning patterns and are better suited towards SFT based distillation from stronger models (which somewhat addresses the stated drawbacks of SFT, if a teacher model is available to produce substantial distillation data)
  - this point is somewhat understandable if it's resource constrained, but the scaling behaviour of this approach still should be further studied

- mitigated impact on instruction-tuned models: the comparison between standard RL and AdaBack on SFT initialized models shows minor differences with slow learning overall. SFT mid/post training pipelines have become fairly standardized in most major model releases, so is the proposition to replace these SFT steps with AdaBack instead?

- reliance on ground truth rationales inherits SFT's data collection drawback

**Questions:**

- How does compute compare across these methods?
  - SFT has the benefit of not needing to sample new sequences, so having a cost controlled comparison might be instructive.

---

> ### Author Response · Authors · 2025-11-20
>
> We thank the reviewer for their positive feedback and detailed assessment of our work. Please find our answers below.
>
> ### Q1. Compute comparison
> RL methods such as GRPO require two steps per iteration:
> 1. sampling multiple candidate solutions per question, and
> 2. updating the model with a value-based objective.
>
> Thus RL is typically more expensive than SFT (where only a forward pass and gradient update are needed.) With KV caching, RL is roughly $2\times$ the cost of SFT for equivalent batch sizes.
>
> For AdaBack, partial solution prefixes reduce the number of tokens generated during rollouts, making each RL iteration more efficient than standard RL.
>
> However, RL and SFT serve different purposes. SFT optimizes likelihood on available demonstrations, while RL aims to improve beyond available demonstrations. Therefore, we note that we cannot merely use SFT instead of RL, mostly due to data limitation. SFT is often used as a predecessor of RL in modern pipelines. Our goal in this paper (and in RL post-training more generally) is to go beyond what is achievable via fine-tuning. As it is reported in our experiments and the literature, RL improves the generalization performance of a model after SFT and goes beyond the generalization that one achieves with SFT.
>
> As also seen in our chain-of-parities experiment, AdaBack surpasses both SFT and standard RL, suggesting that RL (and AdaBack in particular) remains essential for reasoning tasks where training data is limited or insufficiently diverse.
>
>
> ### Q2. The role of SFT: Can AdaBack replace SFT (+RL)?
>
> We appreciate the reviewer’s question. Our findings indicate:
> - When starting from a fine-tuned model, AdaBack consistently improves over standard RL, though the margin is smaller, likely because the model already performs well on the training distribution.
>
> - When starting from a base model, AdaBack sometimes outperforms its application on an SFT-initialized model (Table 1). Our analysis (Section “Does AdaBack Expand the Model’s Solution Space?”) suggests that SFT may reduce exploration entropy, potentially preventing the model from discovering reward-bearing solution modes during RL.
>
>
> This raises a promising—but still open—question of whether AdaBack can fully replace the SFT+RL pipeline. Our current results are mixed, and we believe this requires dedicated large-scale study which we leave to future work. Nevertheless, as mentioned by the reviewer, replacing SFT+RL by AdaBack (without SFT) would increase efficiency and potentially even the generalization performance.
>
> ### W1. Limited scales of the models
> Our experiments are limited to 1B–3B models due to computational constraints. We agree that studying AdaBack at larger scales is important, and recent results (e.g., DeepSeek-R1) suggest that the benefits of RL may become more pronounced as model size increases.
> That said, small- and medium-scale (1B–8B) RL for reasoning remains an active research area, in part because techniques developed at this scale tend to transfer to larger models. Moreover, AdaBack specifically targets the sparse-reward regime of long-horizon reasoning, a difficulty that persists regardless of model size whenever the dataset is sufficiently challenging. For this reason, we expect the qualitative gains of AdaBack to generalize to larger models.
> Please let us know if you have any other concerns.
>
>
> ### References
>
> [1] The Art of Scaling Reinforcement Learning Compute for LLMs, Khatri et al., 2025
>
> [2] POLARIS: A Post-Training Recipe for Scaling Reinforcement Learning on Advanced Reasoning Models, An et al., 2025
>
> [3] AIMO-2 Winning Solution: Building State-of-the-Art Mathematical Reasoning Models with OpenMathReasoning dataset, Moshkov et al., 2025
>
> [4] Open R1: A fully open reproduction of DeepSeek-R1, Hugging Face, 2025

---

### Official Review · Reviewer_HHwF · 2025-11-02

**Soundness:** 3
**Presentation:** 3
**Contribution:** 3
**Rating:** 4
**Confidence:** 4

**Summary:**

This paper proposes AdaBack, an adaptive backtracking algorithm that dynamically adjusts the level of supervision during reinforcement learning training for reasoning tasks. By revealing partial prefixes of target solutions and adapting the supervision ratio based on reward feedback, AdaBack aims to bridge the gap between supervised fine-tuning and RL, enabling models to learn complex reasoning chains more effectively. The authors demonstrate its efficacy on synthetic tasks (e.g., chain-of-parity) and real-world benchmarks (MATH, GSM8k), showing improvements over standard RL and SFT+RL baselines, particularly in out-of-distribution settings.

**Strengths:**

Evaluations span synthetic tasks and real-world reasoning benchmarks, demonstrating broad applicability.
The method is well-motivated and clearly explained, with intuitive visualizations of the training process.

**Weaknesses:**

The approach closely mirrors R3’s reverse curriculum framework, which also uses partial demonstrations to create a curriculum. The distinction—adaptive per-sample scheduling versus fixed stages—is valuable but not thoroughly differentiated in terms of impact.
While outperforming vanilla RL, comparisons to R3 or other curriculum-based RL methods are absent, leaving the reader uncertain about the method’s advantages over existing alternatives.

**Questions:**

None

---

> ### Author Response · Authors · 2025-11-20
>
> Thank you for your review and feedback. Please find our response below.
>
> ### Comparison with R3
>
> AdaBack and R3 share the high-level goal of leveraging partial demonstrations to ease RL training, but they differ substantially in both practical applicability and algorithmic properties.
>
> 1. **Practical challenges of deploying R3**
> R3 requires selecting several dataset-dependent hyperparameters (e.g., number of stages, stage durations, transition rules). For datasets such as MATH or mixed-difficulty collections, these choices are highly ambiguous: It is unclear how many stages are appropriate and how long to train on each stage.
> A short, single-line GSM8K question would be split into the same number of chunks as a multi-step olympiad-level MATH problem, producing mismatched and noisy stage boundaries.
> Further, MATH lacks natural delimiters: reasoning steps often span multiple lines or appear in LaTeX/MathJax environments without clean punctuation, making segmentation brittle.
> Although the R3 paper provides heuristic recommendations, these are dataset-specific and do not generalize to MATH.
> By contrast, AdaBack eliminates the need for any slicing schedule. It requires only a single hyperparameter—the reward threshold—and the curriculum adjusts automatically based on per-sample reward feedback, adapting to both dataset difficulty and intra-dataset heterogeneity.
>
> 2. **Algorithmic and sample-complexity differences**
> We proposed the synthetic chain-of-parities task to compare training methods on problems that require multiple reasoning steps while mitigating the effects of pretraining and data contamination. Here, we demonstrate a separation result:
> There exist problem classes that are not efficiently solvable by RL, SFT, and SFT+RL, but are efficiently solvable (in polynomial time) by AdaBack. Although R3 can make some progress on this task, its lack of adaptivity makes it vastly inefficient by comparison, rendering it unsuitable for long-horizon reasoning tasks.
> This demonstrates that adaptivity at the per-sample level is not merely a heuristic, but can be necessary for solving certain long-horizon reasoning problems. While these theoretical distinctions may not fully manifest on easier datasets, they explain AdaBack’s strong performance as task length or complexity increases.
>
> In short, while R3 uses a global reverse curriculum defined by fixed slices of demonstrations, AdaBack:
> - adapts the supervision length online,
> - requires no predefined segmentation, and
> - leverages each sample’s own reward trajectory.
>
> In settings with many-step reasoning (synthetic or real), this adaptability becomes increasingly important. As shown in Figure 2, R3 and standard RL fail to learn the chain-of-parities task even after 16k iterations, while AdaBack succeeds within 1k iterations.
>
> We are running additional R3 comparisons under matched conditions and will include them as computational resources permit.
>
>
> ### Comparison with other curriculum methods for RL reasoning
>
> We are not aware of other curriculum-learning methods in RL for reasoning that operate in the same regime as AdaBack, aside from R3. The recent works highlighted by Reviewer gh9Y ([1–3]) are valuable, and we will integrate them into our related-work section. However, these approaches are conceptually distinct:
>
> Methods such as AdaRFT, SEC, and RORL modify the batch-level curriculum, i.e., which questions enter each batch.
> AdaBack introduces a per-sample curriculum, i.e., how much of the solution to reveal for each question, while leaving batch sampling unchanged.
> Thus these approaches are orthogonal: AdaBack can be combined with batch-level curricula to potentially further improve training efficiency and generalization.
> If the reviewer has additional curriculum methods in mind that they see as closely related, we would be happy to compare against them.
>
>
> Please let us know if you have any other concerns.
>
> ### References
>
> [1] Shi, Taiwei, et al. "Efficient reinforcement finetuning via adaptive curriculum learning." arXiv preprint arXiv:2504.05520 (2025).
> [2] Chen, Xiaoyin, et al. "Self-Evolving Curriculum for LLM Reasoning." arXiv preprint arXiv:2505.14970 (2025).
> [3] Bae, Sanghwan, et al. "Online difficulty filtering for reasoning oriented reinforcement learning." arXiv preprint arXiv:2504.03380 (2025).

---

> > ### Comment · Reviewer_HHwF · 2025-11-27
> >
> > I have read the response, and I will keep the score.

---

### Official Review · Reviewer_gh9Y · 2025-11-02

**Soundness:** 2
**Presentation:** 3
**Contribution:** 2
**Rating:** 4
**Confidence:** 4

**Summary:**

This paper introduces AdaBack, an adaptive curriculum method for reinforcement learning that dynamically reveals partial solution prefixes (rationales) based on performance feedback. For each example, AdaBack maintains a supervision ratio $\rho$ that determines the proportion of the solution prefix shown to the model. The model is then trained using standard RL with a verifiable reward on these partial solutions. The supervision ratio is updated via a stochastic “binary search” around a reward threshold. AdaBack aims to bridge the gap between supervised fine-tuning (dense feedback) and reinforcement learning (sparse feedback) by maintaining a non-vanishing reward rate. Experiments on MATH, GSM8k, and the symbolic manipulated variants of GSM8k demonstrate that AdaBack outperforms GRPO, especially on tasks with sparse rewards.

**Strengths:**

- The proposed method improves model performance on mathematical datasets. In particular, it demonstrates greater benefits on the Base-7 and Tensor-2 GSM8k datasets—variants of GSM8k created by symbolic manipulations and effectively reduce the density of reward signals during training. These results align with the goal of addressing the sparse reward problem.

- The paper includes and analyzes negative results (e.g., saturation on MATH for Llama‑3.2‑3B‑Instruct and Qwen‑2.5), which increases the paper’s value for future work.

- The proposed method is well-motivated and clear-explained.

**Weaknesses:**

- The paper does not specify how many random seeds were used to compute test accuracy. Given the high randomness in LM sampling, this omission raises questions about statistical robustness, especially for Table 1.
- The method learns a per-problem parameter $\rho$ and initializes it using a global EMA with hyperparameter $\alpha$, but the choice and sensitivity of $\alpha$ are not discussed.
- Although the approach is inspired by R3, no baseline comparison with R3 is included on the mathematical datasets, leaving open how much of the improvement derives from the adaptive curriculum itself.
- The experiments are limited to Llama models. Prior studies indicate that model families like Qwen behave differently in RL fine-tuning. Thus, the generality of AdaBack’s gains remains uncertain.
- The method requires on ground-truth rationales, which constrains scalability and comparability to pure RL approaches that rely only on verifiers.
- (Minor issue) Some recent work on curriculum learning in RL fine-tuning, such as AdaRFT (Shi et al., 2025), SEC (Chen et al., 2025), and RORL (Bae et al., 2025), is not discussed in the related work section.

Shi, Taiwei, et al. "Efficient reinforcement finetuning via adaptive curriculum learning." arXiv preprint arXiv:2504.05520 (2025).

Chen, Xiaoyin, et al. "Self-Evolving Curriculum for LLM Reasoning." arXiv preprint arXiv:2505.14970 (2025).

Bae, Sanghwan, et al. "Online difficulty filtering for reasoning oriented reinforcement learning." arXiv preprint arXiv:2504.03380 (2025).

**Questions:**

- How many random seeds were used for the accuracy results in Table 1?
- What value of $\alpha$ was used for the global average initialization, and how sensitive is final performance to this choice?
- What is R3’s performance on the MATH and GSM8k datasets under the same setup?
- Have you tested AdaBack on other model families, such as Qwen, to evaluate its generalization? I understand that Qwen may have saturated the MATH dataset, but there are other, more challenging datasets where a dense reward could still be beneficial.

---

> ### Author Response · Authors · 2025-11-20
>
> We thank the reviewer for the detailed and constructive feedback. We address each point below.
>
> ### W1, Q1. Number of random seeds
> We understood that your question is on LLM sampling, not restarting all the experiments with a few different seeds. Otherwise, running full RL training runs with multiple seeds is computationally prohibitive (four additional seeds roughly quadruples the total compute, taking us from ~80k GPU hours to 400k.)
> Instead of running on multiple seeds, we report accuracy and reward by averaging over the last 5 checkpoints of each run (discussed in the paper in line 931). This procedure averages over two sources of randomness:
>  (i) stochastic sampling during generation, and
>  (ii) the variability across different points in the RL trajectory.
>  This yields a stable estimate without incurring the cost of full multi-seed retraining.
>
>
> ### W2, Q2. Initialization and sensitivity of the moving averages
>
> For each problem, AdaBack maintains lower and upper bounds on the supervision ratio interval. These are updated using an exponential moving average (EMA)
>
> $$\text{EMA}^{(t+1)} = \alpha \text{EMA}^{(t)} + (1-\alpha) (\text{new observation}),$$
>
> with $\alpha = 0.8$, lower bound initialized to $0$, and upper bound to $0.9$.
>
> We experimented with several initializations and a range of $\alpha$ values. Across all settings, we observed no meaningful differences in final performance. This is because the EMAs converge rapidly early in training, as also visible in Figure 2 (left), where the average supervision ratio quickly stabilizes. Thus the method is robust to these choices. Thanks for bringing this gap to our attention. We will include this in the discussion of the moving average after line 909.
>
> ### Q3. R3’s performance on MATH and GSM8K
>
> We acknowledge that including R3 results on MATH and GSM8K would strengthen the empirical comparison and are running these experiments.
>
> However, applying R3 to these datasets is non-trivial for two reasons:
>
> 1. Dataset-specific slicing challenges: R3 requires demonstrations to be segmented using a suitable delimiter.
> In particular,  MATH lacks consistent delimiters: many steps span multiple lines, contain MathJax blocks, or omit punctuation, making segmentation ambiguous. Choices such as period-based splitting or uniform slicing (e.g., 5–6 chunks) lead to inconsistent stage definitions across heterogeneous datasets.
> These issues motivated AdaBack’s per-sample adaptive approach, which eliminates the need for dataset-specific slicing heuristics.
>
> 2. R3’s reported GSM8K performance is below current baselines:
> R3’s original GSM8K results are on Llama-2 7b (SFT: 41.6 → RL: 44.7 → R3: 50.5) and they fall below baselines reported elsewhere (e.g., LLaMA-2 7B achieves ~49.5% zero-shot and >56% with 5-shot prompting, see for instance arXiv:2403.04706). Matching or surpassing these baselines required extensive retuning of R3 work, making a fair comparison difficult under reasonable compute budgets.
>
> Given these challenges, our comparisons focused on the chain-of-parities experiment, where properties of the curriculum can be isolated and controlled. In this setting, AdaBack succeeds in <1000 iterations (~2 hours on one node), whereas both R3 and standard RL fail to learn even after 16,000 iterations (>1 day on one node).
>
> Nevertheless, we are continuing to run additional experiments under matched conditions for inclusion in an updated version; we hope to be able to deliver the numbers before the end of the rebuttal period.
>
>
>
>
> ### Q4. Experiments on the Qwen model family
>
> We ran preliminary experiments on Qwen2.5-1.5B (base and instruct). Consistent with line 409, these models saturate quickly on MATH, similar to LLaMA-3.2B-Instruct. We will add corresponding plot for Qwen in the revision (very similar learning curves to Figure 8: Lack of Learning Signal on MATH).
> Evaluating AdaBack on more difficult datasets with Qwen models would indeed be interesting. However, large-scale reasoning datasets with clean human-written rationales are limited. Most of the available datasets (e.g., OpenR1-Math-220k) contain LLM-generated rationales, many of which are noisy or incorrect, and are 10 times larger than MATH/GSM8K, requiring substantially higher compute (see [1,2,3]).
> More broadly, AdaBack addresses a model-agnostic sparse-reward exploration problem. We therefore expect its benefits to transfer to other model families when training on sufficiently challenging datasets.

---

> > ### Author Response · Authors · 2025-11-20
> >
> > ### W6. Additional related work
> > We appreciate the reviewer highlighting recent curriculum-learning approaches (AdaRFT, SEC, RORL) and will integrate them into the related-work section.
> > These methods primarily adjust dataset/batch-level curricula, i.e., selecting which questions to include in each RL batch. By contrast, AdaBack introduces a per-sample curriculum, adaptively revealing partial rationales while sampling questions uniformly. Thus these approaches are complementary: AdaBack can be combined with batch-level curricula but unlike these works which do not benefit from chain-of-thought (CoT) demonstrations, AdaBack uses CoT supervised data to further improve efficiency.
> >
> > Please let us know if any additional clarifications would be helpful.
> >
> > ### References
> >
> > [1] POLARIS: A Post-Training Recipe for Scaling Reinforcement Learning on Advanced Reasoning Models, An et al., 2025).
> > [2] AIMO-2 Winning Solution: Building State-of-the-Art Mathematical Reasoning Models with OpenMathReasoning dataset, Moshkov et al., 2025
> > [3] Open R1: A fully open reproduction of DeepSeek-R1, Hugging Face, 2025)

---

> ### Author Response · Authors · 2025-11-27
>
> Dear Reviewer,
>
> As discussed in our response to Q4, we have added Qwen2.5-1.5B (base and instruct) learning curves on GSM8K dataset.  These plots, which appear on the final page of the updated appendix in the current pdf, show that both models saturate after only a few iterations which mirrors the behavior we observed for Llama-3.2-3B-Instruct in Figure 8. This confirms that, for this model family, our datasets provide very limited learning signal for RL methods beyond what the model has seen at pretraining.
>
> Please let us know if any further details would be helpful.

---

### Author Response · Authors · 2025-12-03
**Summary and Response to AC**

Dear AC and reviewers,

We thank all reviewers for their thoughtful evaluations. We have spent significant effort addressing their comments, particularly by running several new experiments and incorporating a new dataset (DeepScaleR). We are encouraged that reviewers found the paper clearly written and the contribution, a per-sample adaptive curriculum between SFT and RL, well-motivated.

In the revised version uploaded during the rebuttal and in our comments, we incorporated the following key additions:

### 1. Comprehensive Comparison with R3 (Appendix G)

As requested, we conducted a direct comparison between AdaBack and R3 on three datasets of increasing difficulty: GSM8K < MATH < DeepScaleR, using Llama-3 1B and 3B. Across all settings, both methods use identical ground truth, rewards, and models; the only difference is the curriculum mechanism.
We observe a trend aligned with difficulty. AdaBack outperforms R3 on DeepScaleR (both sizes) and on MATH (1B model). It matches R3 on MATH (3B) and performs slightly below R3 on the easier GSM8K. The table of results and further explanation can be found in Appendix G, Table 2, in the revised manuscript.

**Advantage of AdaBack on Long-Horizon Tasks.** We draw attention to our chain-of-parities experiment (Figure 2), which shows that as the number of reasoning steps increases, AdaBack becomes >20× more efficient than R3. As the real-world datasets we considered in this paper still have only a couple of reasoning steps, we believe the chain-of-parties experiment better reflects AdaBack’s advantages compared to R3. AdaBack is uniquely suited for long-horizon tasks (e.g., math Olympiad problems), though data acquisition currently limits large-scale training in that regime.

### 2. Additional Experiments & Clarifications

- Qwen2.5 family: We added new experimental results for Qwen2.5-1.5B (Base/Instruct) on GSM8K (Appendix G). The learning curves show saturation similar to Llama-3 Instruct model, confirming that standard datasets offer limited additional RL signal for these strong base models.

- Reward threshold sensitivity: In response to Reviewer hbbC, we added a sensitivity study (Appendix G). AdaBack’s performance remains stable across a broad range of threshold values, with only extreme values slightly affecting early training.

- Clarifications on positioning of the paper: We clarified that AdaBack serves as a drop-in replacement for standard SFT+RL (using the same data), and expanded related work to discuss batch-level curricula (AdaRFT, SEC, RORL) as complementary to our per-sample approach. We also clarified the exponential moving-average update of the supervision interval and reported that performance is insensitive to the initialization and EMA parameter in our experiments.

We hope these additions address the concerns raised, and we are grateful for the feedback that helped strengthen the paper.

---

### Meta-Review · Area_Chair_je2A · 2025-12-24

**Summary:**

This paper introduces AdaBack, an adaptive backtracking algorithm designed to bridge the gap between supervised fine-tuning and reinforcement learning for sequence generation tasks. By dynamically revealing partial solution prefixes based on per-sample performance feedback, the method creates an incremental curriculum that helps models navigate sparse reward landscapes. Experiments on synthetic parity tasks and mathematical reasoning benchmarks demonstrate that AdaBack enables models to acquire new reasoning capabilities, outperforming baselines in sparse reward scenarios.

**Strengths:**
1. The proposed AdaBack method offers a principled and novel approach by implementing per-sample adaptive partial supervision rather than relying on global heuristic schedules. This mechanism effectively addresses the sparse reward problem in complex reasoning tasks by dynamically scaling difficulty based on individual model performance.
2. Empirical evaluations are comprehensive, spanning both synthetic latent parity tasks and real-world mathematical benchmarks like MATH and GSM8k. The results demonstrate significant improvements, especially in sparse reward settings like Base-7 and Tensor-2 GSM8k variants.
3. The paper provides high-quality analysis, negative results and clear visualizations of training dynamics, which help interpret how the method expands the solution distribution.

**Weaknesses:**
1. Comparisons with related methods like R3, AdaRFT, and SEC are absent, leaving uncertainty about AdaBack's advantages over existing curriculum-based RL approaches.
2. Since the approach relies on ground-truth rationales to provide partial supervision, it inherits the expensive data collection constraints of supervised fine-tuning and lacks the scalability of pure reinforcement learning.
3. The scope of the study is primarily limited to smaller model sizes and relatively simple benchmarks, making it difficult to assess how the method scales to larger architectures or more difficult tasks.
4. There is a lack of systematic sensitivity analysis for key hyperparameters, which is necessary to understand the method's stability across different domains.

**Reviewer Concerns:**

In the revised version, the authors have incorporated the following key additions: (1) A direct comparison between AdaBack and R3 was conducted across three datasets. AdaBack outperforms R3 on DeepScaleR (both sizes) and on MATH (1B model), matches R3 on MATH (3B), and performs slightly lower than R3 on the easier GSM8K dataset. (2) New experimental results for Qwen2.5-1.5B (Base/Instruct) on GSM8K are added in Appendix G; the learning curves exhibit saturation characteristics similar to those of the Llama-3 Instruct model. (3) A sensitivity study has been added to the revised version, which shows AdaBack’s performance remains stable across a broad range of threshold values.

**Reviewer Scores:**

Reviewer HHwF stated that he/she would maintain the score. However, despite assigning an overall rating of 4, he/she rated the paper’s Soundness, Presentation, and Contribution all as "good". Reviewer hbbC expressed gratitude for the authors’ detailed responses and thus may raise the score.

---

### Decision · Program_Chairs · 2026-01-26

Accept (Poster)